# How to steer active colloids up a vertical wall

**Adérito Fins Carreira[1,4], Adam Wysocki[2,4], Christophe Ybert ©[1], Mathieu Leocmach ©[1], Heiko Rieger ©[2,3] ✉ & Cécile Cottin-Bizonne ©[1] ✉**

An important challenge in active matter lies in harnessing useful global work from entities that produce work locally, e.g., via self-propulsion. We investigate here the active matter version of a classical capillary rise effect, by considering a non-phase separated sediment of self-propelled Janus colloids in contact with a vertical wall. We provide experimental evidence of an unexpected and dynamic adsorption layer at the wall. Additionally, we develop a complementary numerical model that recapitulates the experimental observations. We show that an adhesive and aligning wall enhances the pre-existing polarity heterogeneity within the bulk, enabling polar active particles to climb up a wall against gravity, effectively powering a global flux. Such steady-state flux has no equivalent in a passive wetting layer.

Self-propelled agents, like bacteria, cells, micro-organisms, animals, or synthetic active particles, inject energy at small scale into their environment, driving the system out of equilibrium. Often, this energy only powers disordered agitation akin to thermal energy[1] and a major challenge is to better understand active energy flows to extract useful work from seemingly low-grade heat. This was achieved in a few configurations with, for instance, swimming micro-organisms driving unidirectional rotation[2], or decreasing the apparent viscosity of their sheared medium[3], both effects that have no equivalent in equilibrium systems. To date, such active energy harvesting has not been exploited in the ubiquitous configuration of self-propelled particles exposed to a constant and uniform force, e.g., gravity. There, so far, active sedimentation properties have been shown to be roughly that of a hotter suspension[4] although finer examination reveals intrinsic out-of-equilibrium characteristics[5,6]. In this paper we address the question: Does polar active matter exhibit capillary rise? To investigate this question we introduce a lateral wall in contact with sedimenting polar active particles, a configuration which may fall within active wetting phenomena.

A straightforward mean for harvesting active self-propulsion forces is to herd the system. This can be obtained with a polarizing external field, where particles align with the field and are harnessed to produce useful work. For instance, the swimming direction of magnetotactic bacteria can be polarized by a magnetic field, producing a net flux[7]. Bottom-heavy algae[8], colloids[9–11], or walkers[12] may point up, effectively acting against gravity. Yet, in the absence of external torque acting on particles, a constant and uniform force field can still promote

a local averaged polarization[5,6]. However, this polarization only arises self-consistently to balance the sedimenting flux such that no net flow is created at steady-state.

Confining walls are known to promote complex responses of active systems which challenge the intuition one has for the equilibrium[13]. At odds with at-equilibrium thermal motion, the pressure exerted by self-propelled particles on walls depends on the wall stiffness[14] and curvature[15,16]. Alike, because of directional persistence[17,18], self-propelled particles tend to accumulate at and to polarize towards walls. In addition, intrinsic wall-particle interaction, in particular aligning interactions, have a major influence on the detention time of the particles[18].

Considering a heterogeneous active system displaying a fluid-fluid interface, the introduction of a wall extends wetting phenomena to active systems. With simple molecular liquids, surface tension that stems from the attractive interactions of the molecules with the wall and between each other[19] can trigger "wall-climbing": from the formation of a wetting layer, to a meniscus, to capillary rise. Yet, the steady-state of this equilibrium system displays no net flux. Such phenomenology also extends to complex fluids such as a colloidal suspension with attractive interactions[20,21]. In this case the interfacial tension between the colloid-rich and the colloid-poor phase is ultralow leading to mesoscopic interface fluctuations. Starting from such equilibrium wetting configuration of phase separated polymer solutions near a vertical wall, a recent experimental and theoretical study[22] demonstrated surprising wetting-like phenomena when adding an active nematic into one of the two phases. In particular, an activity-induced transition towards full wetting was observed.

[1]Université de Lyon, Université Claude Bernard Lyon 1, CNRS, Institut Lumière Matière, Villeurbanne, France. [2]Department of Theoretical Physics and Center for Biophysics, Saarland University, Saarbrücken, Germany. [3]Leibniz Institute for New Materials INM, Saarbrücken, Germany. [4]These authors contributed equally: Adérito Fins Carreira, Adam Wysocki. ✉e-mail: heiko.rieger@uni-saarland.de; cecile.cottin-bizonne@univ-lyon1.fr

Even without attractive interactions, active particles can also display a purely out-of-equilibrium separation into a dense and a dilute phase, called motility induced phase separation (MIPS), which is due to a slowdown occurring during collisions between particles[23]. We may thus anticipate for such repulsive active systems a possible extension of the passive scenario, although the mere definition of surface tension is difficult and may yield negative values[24,25], which should be incompatible with a stable interface. Indeed, a recent theoretical study demonstrates such an analogue out-of-equilibrium behaviour, where a repulsive active dense phase can form a meniscus on a wall or rise against gravity in a confining channel due to the slowdown that occurs during particle-wall collisions[26].

Here, we present experimental observations of the behaviour of a non phase separating assembly of active colloids that are sedimenting in a gravitational field in the presence of a lateral wall and explore the question of active capillary rise for polar active matter. We provide the evidence that even without phase separation polar active matter can display an unexpected and dynamic adsorption layer at the wall that rises against gravity and has no analogue in passive systems. We also offer a numerical model that recapitulates the experimental observations. It enables us to understand the physical mechanisms at the origin of the observations and to test the respective influences of adhesive and aligning interactions with the wall, parameters that are challenging to adjust experimentally. Our results demonstrate that a lateral wall can act as a pump against a force parallel to it, opening the door to active microfluidic circuits where a configuration as simple as gravity and walls could play a role analog to a generator in an electric circuit.

## Results

### Experimental and numerical observation of "Wetting-like" behaviour

We study experimentally the behaviour of sedimenting active colloidal particles in the presence of a vertical wall. The experimental configuration, as shown in Fig. 1 and explained in detail in the Methods section, is composed of Janus microspheres of average radius $R = 0.8$ μm, sedimenting along the vertical direction $z$ under an effective gravity $\vec{g}^*$. Those particles self-propel through phoretic effects in the presence of hydrogen peroxide[27,28]. We quantify the activity of the system with the swim Péclet number $Pe_s$ that we measure by analyzing the sedimentation profile, in the dilute regime, far away from the wall, assuming a Boltzmann distribution[28,29] (see Methods). This $Pe_s$ indicates how far the system is out-of-equilibrium. By changing the concentration of hydrogen peroxide, $Pe_s$ can typically be tuned between 0 and 14.

A wall is introduced in the system as a glass capillary oriented along $z$ and immersed down to the colloidal sediment. To study how this wall impacts the active particles system, we measure the density field $\phi(x, z)$ for various $Pe_s$. In Fig. 1 we show images of the sediment together with a few iso-densities (see Methods) in the passive and active cases. As expected for purely repulsive particles with no wall-attraction, passive colloids at $Pe_s = 0$ (Fig. 1A, B) are hardly affected by the wall[30] and the passive system does not adsorb at the wall.

As we activate the colloids with hydrogen peroxide ($Pe_s > 0$), iso-densities remain horizontal and parallel to each other far from the wall. This is what is expected in the simplistic picture where activity results in a hotter system with a higher sedimentation length (Fig. 1C, D). Nevertheless, this simplistic hot-colloids mapping strikingly breaks down in the vicinity of the wall, where we evidence an upturn of the iso-densities that rise with increasing activity. The wall causes particles to climb up at its contact. Indeed, we observe what seems to be a wetting or an adsorption layer, a layer of one particle thickness. Past this wall region, we note a small dip of the iso-densities before progressively returning to the far-field "flat" region. Overall, this departs strongly from the global features of classical passive wetting or of recent

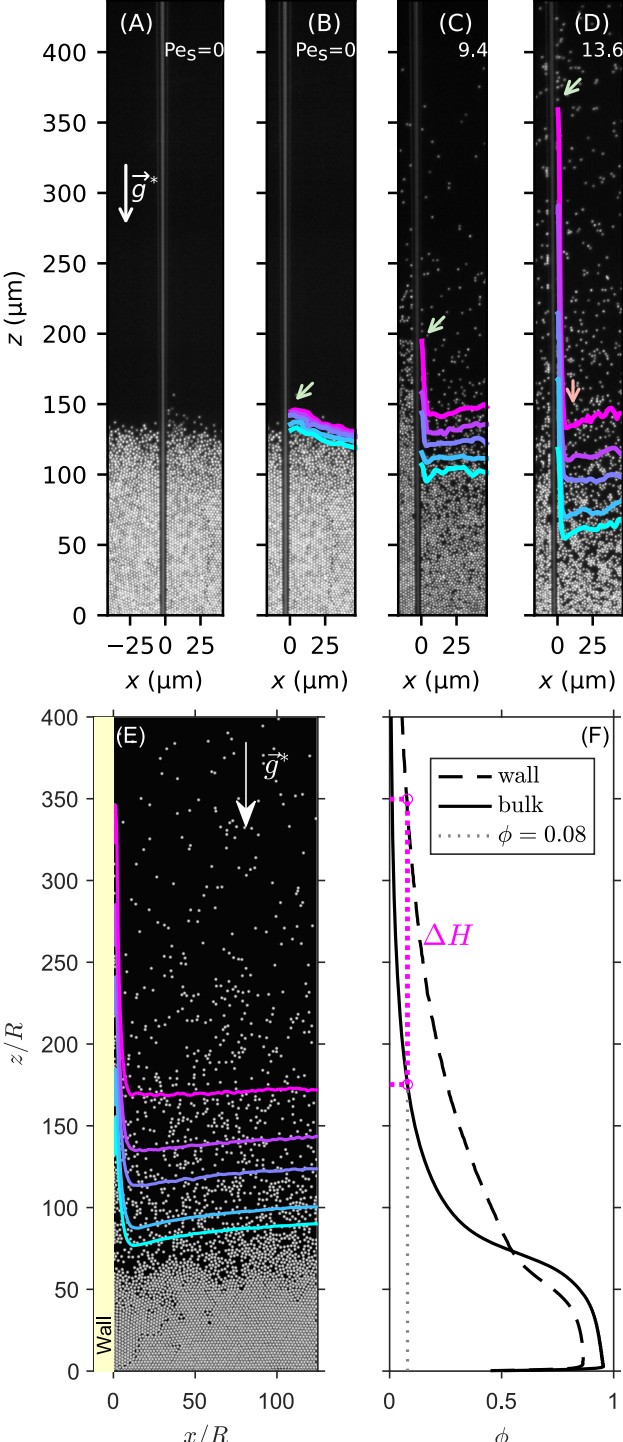

**Fig. 1 | Active Janus colloids climb a wall. A** Experimental set-up: glass wall dipped into a sedimented monolayer of passive colloidal particles, at $Pe_s = 0$, under a gravity $\vec{g}^*$. **B**–**D** Iso-density maps of the colloids at $Pe_s = 0$ (passive case) and for two activities $Pe_s = 9.4$ and $Pe_s = 13.6$. The iso-density values $\phi$ are from top to bottom 0.08, 0.12, 0.16, 0.24, and 0.3. The pale green arrow highlights the maximum height of the isodensity curve $\phi = 0.08$ at the adsorption layer. The pale red arrow indicates the small depression of the density close to the wall. **E** Snapshot of the numerical simulation of an assembly of ABPs under gravity near a wall with alignment $\bar{l} = 26$ and adhesion strength $\bar{\epsilon} = 3.25$, respectively, at an activity $Pe_s = 13$ (see Methods for details). Isolines at densities $\phi = 0.08, 0.12, 0.16, 0.24, 0.3$ are shown (from top to bottom). **F** Density profiles $\phi(z)$ predicted by our ABP model far from the wall (solid line), and in the *adsorption layer* (dashed line). The dotted line indicates the density $\phi = 0.08$ from which the adsorption layer height $\Delta H$ is defined.

extension with active perturbation[19,22], where a macroscopic upward meniscus smoothly bridges between the wall and far-field unperturbed region. Now that we have evidenced "wall-climbing" against gravity in our polar active system, we will in the following quantify this rise and reveal the mechanism leading to it.

In a classical picture, colloids excess at the wall comes from attractive wall-particle interactions. Indeed, as already mentioned in the introduction, wall accumulation of active particles is a generic feature for which directional persistence provides an underlying mechanism for *effective* adhesion[17,29]. However, in specific systems, interactions between walls and active-particles comprise a rich variety of contributions. For instance a *direct* wall-particle attraction may also arise due to activity through phoretic or hydrodynamic effects[29,31,32]. Likewise, hydrodynamic interaction of dipole micro-swimmers with surfaces can induce alignment with the wall[31,33–35].

To complement our experimental evidences, we also analyzed the predictions of a model of sedimenting repulsive active Brownian particles (ABPs)[36] in the presence of a vertical wall. As we shall discuss, this allows us to explore the importance of the detailed interaction between walls and active-particles. When including an activity-induced interaction, mimicking known diffusiophoretic adhesion[29] as well as a wall bipolar (also called nematic) alignment parallel to the wall[33,34,37] mimicking hydrodynamic torque (see Methods for details), the whole experimental phenomenology is reproduced. As is shown in Fig. 1E, numerical simulations display both the adsorption layer at the wall in the active case and the small dip close to the wall. Note that consistently with experiments, we explored here an activity region for which no MIPS occurs, corresponding to moderate activities quantified by $Pe_s \le 17$, far enough from $Pe_{crit} \approx 26.7$[38] (see Methods for the definition of the normalized swim persistence length $Pe_s$). In the MIPS regime self-propelled particles generate a phenomenology that is reminiscent of classical wetting configuration as a recent theoretical study predicted a smooth macroscopic wetting meniscus[26], in contrast with the present observations. Therefore, the layer we observe is not wetting layer per say (implying a gas-liquid phase separation), but an adsorption layer. In the following we investigate the physical origin of this adsorption, how it interacts with gravity and how this phenomenology is specific to polar active matter and has no equilibrium analogue.

## Detention time at a wall without gravity
Before interpreting our results, we have to understand how active particles accumulate at a wall in the absence of gravity and the influence of wall-particle interactions on this accumulation. To do this, we generalize arguments that have been laid out in the case of steric[17] and hydrodynamic interactions[18]. To simplify the argumentation, we neglect the translational diffusion, i.e. $Pe_s \gg 1$. In the absence of attractive interactions, an active particle is able to escape the wall as soon as the normal component of its propulsion force points away from the wall. Since its orientation evolves through rotational diffusion, a particle polarized towards the wall will have a long detention time, whereas a particle polarized tangentially to the wall will have a shorter detention time. Therefore, wall-aligning interactions actually reduce detention time because they bring the particles closer to the escape angle. By contrast, attractive interactions push the escape angle away from the wall and thus increase detention time. A combination of strong attractive and strong wall-aligning interactions traps the orientation parallel to the wall while pushing the escape angle away, and thus increases detention time. However, the respective magnitudes of wall-attraction and wall-alignment might affect detention time in a more nuanced way. Let us now explore how a force parallel to the wall affects these behaviours and how it explains our observation of an activity-induced adsorption layer. To do this, we use our ABP model to explore systematically the influence of the above-mentioned contributions, namely self-propulsion, direct wall adhesion, wall alignment, and the combination of these.

## Adsorption layer height: passive adhesive versus active adhesive
From the density field, we obtain the density profiles $\phi_{wall}(z)$ close to the wall (within the adsorption layer) and $\phi_{bulk}(z)$ in the bulk, far away from the wall (see Methods). It is clear from Fig. 1F that at a given altitude, the adsorption layer has an excess density as compared to the bulk and that the altitude corresponding to a certain density is much higher in the adsorption layer than far from the wall. The height difference between the adsorption layer and the bulk gets even larger as the density decreases. We now define the *adsorption layer height* $\Delta H$ as the difference in altitudes corresponding to a given density of $\phi = 0.08$ in the adsorption layer and the bulk. $\Delta H$ is a global observable of the activity-induced phenomenon.

For a passive system, wall adhesion promotes the creation of a gravity-fighting adsorption layer[20,30,39], as can be seen in the inset of Fig. 2A where however $\Delta H$ only reaches a few particle radii $R$. Now, let us consider an active system experiencing both wall accumulation due to the persistence of motion and an activity-induced *direct* adhesion. Indeed, for Janus colloids, the self-generated electro-chemical gradients responsible for self-propulsion also generate a wall-attraction force, thus scaling as the propulsion velocity (see Supp. Mat. for an explanation for the form of the adhesive energy, along with simple estimation on its magnitude). Accordingly, we take a wall adhesion strength as $\tilde{\epsilon} \propto Pe_s$ in our ABP model. Comparing passive and active systems with same direct adhesion parameter in the inset of Fig. 2A, we observe that $\Delta H$ of self-propelled particles exceeds by more than a decade the one of passive colloids. This is a further illustration that this active system does not merely behave as an equilibrium system regarding adsorption properties, with self-propulsion of individual particles a key factor of the global response.

## Péclet number dependency
In Fig. 2A, we thus focus on self-propelled particles showing $\Delta H$ obtained numerically for different levels of activity as quantified by Péclet number $Pe_s$. A neutral wall represents the benchmark situation where no direct interaction —neither attraction nor orientation— is present aside the short-range steric repulsion. We observe in this case that pure activity, on a neutral wall, generates an adhesion layer with $\Delta H$ increasing with $Pe_s$. Activity has thus an effect that mimics adhesion. In line with above discussion, adding a *direct* activity-dependent wall adhesion increases the adsorption layer height as compared to neutral wall.

To mimic the tangential alignment of Janus colloids with a wall, we choose an alignment strength as $\tilde{\Gamma} \propto Pe_s$. This is consistent with the experimental characterization of Janus colloids as force dipole microswimmers[40] which strength is proportional to the propulsion speed[34] (see Supp. Mat. for an explanation of the form of the alignment strength, along with the simple estimation of its magnitude). We observe that pure wall alignment interaction decreases $\Delta H$ as compared to the neutral case. So far, the prediction of our ABP model is consistent with the influence of wall-particle interactions on detention time without gravity, as if the detention time drives the adhesion height. Indeed in presence of gravity we can directly measure the detention time of a single particle $\tau_{detention}$, that is the average time a particle remains in the adsorption layer (Fig. 2B). Our ABP model predicts how $\tau_{detention}$ depends on the particle-wall interaction. As expected, adhesion always increases detention time as compared to a neutral wall. On the other hand, particles escape a purely aligning wall sooner with increasing activity.

As shown also in (Fig. 2A, B), we were also able to measure both $\Delta H$ and $\tau_{detention}$ in experiments. There, we observe an increase of $\Delta H$ with activity, with higher values than neutral wall case, and a small decrease of $\tau_{detention}$ with activity. Such trends are not compatible with a pure wall-alignment or a pure wall-adhesion. To explain our experimental observations, we have explored the combined effect of adhesion and alignment. We have used our ABP model to construct a map of the detention time function of wall-attraction $\tilde{\epsilon}$ and wall-alignment $\tilde{\Gamma}$ for

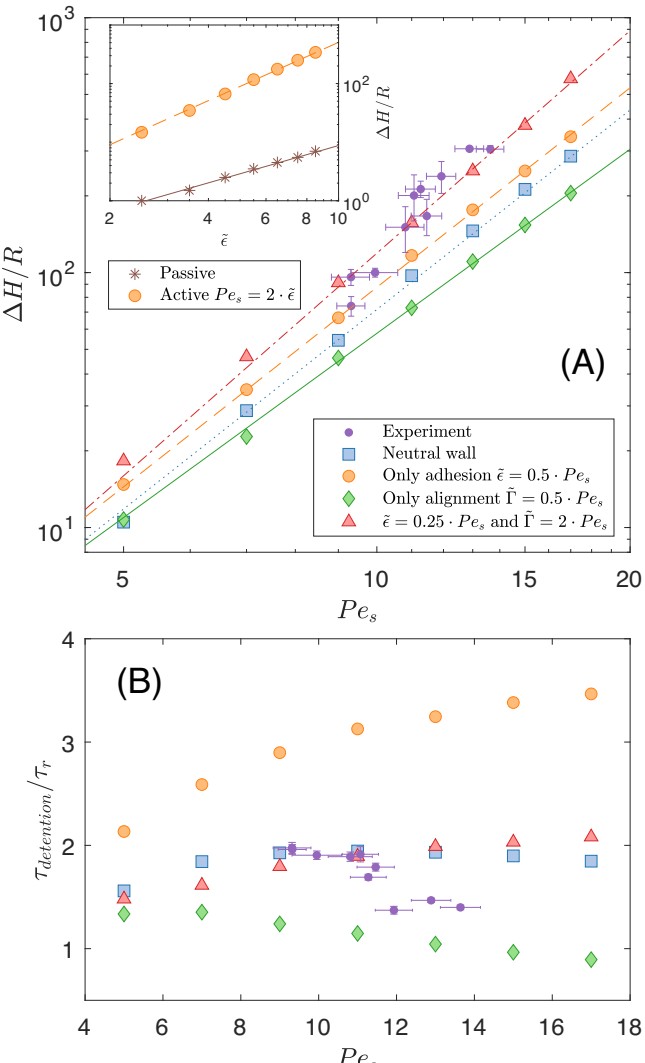

**Fig. 2 | Adsorption layer height dependency. A** Numerical values of the adsorption layer height $\Delta H$ as a function of the wall adhesion strength $\tilde{\epsilon}$ in the passive (brown stars) and active (orange dots) case, where in the latter case we assume that the adhesion strength increases linearly with activity as $\tilde{\epsilon} = 0.5 Pe_s$. Straight lines are best power-law fits. **A** Adsorption layer height $\Delta H$ as a function of the activity $Pe_s$ from experiment (purple dots) and from simulations, for, a neutral wall (blue square), a purely adhesive wall (orange circle) with $\tilde{\epsilon} = 0.5 Pe_s$, a purely aligning wall (green diamond) with $\bar{\Gamma} = 0.5 Pe_s$ and a wall with both alignment and adhesion (red triangles) with $\tilde{\epsilon} = 0.25 Pe_s$ and $\bar{\Gamma} = 2 Pe_s$. Straight lines are best power-law fits. **B** Detention time of colloids without a neighbour in the *adsorption layer* versus the activity $Pe_s$.

$Pe_s = 13$ (see Supplementary Fig. 6) and found a good match, within our model, to the experimental data for $\tilde{\epsilon} = 0.25 Pe_s$ and $\bar{\Gamma} = 2 Pe_s$. The Péclet dependencies with these parameters are shown in red in (Fig. 2A, B). We observe a good agreement for $\Delta H$ between the experiments and simulations. The detention time values are similar, however, experiments suggest a weak decreasing trend that is not captured by the simulations. Achieving a more accurate agreement would require to refine the ABP model, particularly hydrodynamic interactions with the wall are missing. Also, the chemical gradient around the particle is modified by the presence of the wall, which should influence the velocity of the particle as well. A full description should involve taking into account both those hydrodynamics and diffusiophoretic effects and their coupling[31,35,41,42].

Furthermore with this choice of parameters, although the detention time is similar to its numerical values in the neutral wall case, the adhesion height is much larger. This is inconsistent with the simple picture that $\tau_{detention}$ drives $\Delta H$. Indeed in the following, we shall see that part of the adsorption layer height is due to the singular influence that a wall parallel to gravity exerts on polarity and fluxes.

## Polarization at the wall

To explain the adsorption layer height, we now consider the defining property of polar active matter, i.e. polarity. As we already mentioned, a remarkable feature of sedimented active particles is the existence, in the absence of lateral walls, of a non-vanishing local polarization **M(r)**, and in particular a non-zero component parallel to gravity $M_z$[5,6]. In the dilute regime, the mean orientation points upward ($M_z > 0$) to balance the downward sedimentation and to guarantee the absence of a net particle flux. By contrast, in the dense sediment at the bottom of the cell there is a downward polarization ($M_z < 0$), as the total polarization has to vanish[43]. In our ABP model, this bulk behaviour is well recovered far away from the wall. Indeed, Fig. 3A shows in pink the polarity distribution in the dilute bulk as a circle slightly shifted upward. We found that this polarity distribution far from the wall is independent of wall-particle interactions.

Let us now focus on polarity distribution in the adsorption layer. The measure of polarity in the Janus colloidal particles, required a specific experiment. The method, fully detailed in Supp. Mat., is based on the tiny shifts between a particle location in the three color channels of a camera that correspond to the colour difference between the two faces of the Janus. As chromatic aberrations are benchmarked in the bulk region, only excess polarization at the wall is reported. In Fig. 3A, we compare this experimental excess polarization to absolute numerical polarisation distributions for different wall interactions. In all cases, we observe a net polarization upward as in the bulk, but also a net polarization towards the wall. Indeed, as recalled above in the absence of gravity, a particle pointing towards the wall will on average remain longer at the wall than a particle nearly parallel to it, or worse pointing away from the wall.

Furthermore, when wall-particle bipolar alignment interaction is present, the polarization distribution is elongated parallel to the wall. Our experimental distribution, although noisy, displays strongly this elongation, which is a clear and independent indication of alignment interaction in the experiment, compatible with our choice of alignment strength $\bar{\Gamma} = 2 Pe_s$.

More quantitatively, in Fig. 3B we report numerical time- and ensemble-averaged vertical polarization profiles $M_z(z)$. In all cases, the same qualitative picture is retained, with the averaged polarization reversing from an upward to a downward orientation when going from large $z$ down into the sediment. At the wall, however, the polarization strength becomes quantitatively dependent on the wall-particle interaction properties. For a neutral or a purely adhesive wall, the polarization at the wall is lower than in the bulk. Indeed, as recalled above, detention effects give higher probability to wall-facing orientations at the expense of tangential and outward ones. Therefore vertical non-aligning walls will act as dampers for the bulk $z$-polarization. On the opposite, a vertical bipolar aligning wall will boost any initial upward or downward polarization bias in the particles orientation. This is what we observe (Fig. 3B) for all aligning configurations (with or without additional adhesion) where the wall acts as an enhancer of the bulk polarization.

## Fluxes, circulations, and wall dynamics

As we just pointed out, the colloid polarization at a vertical wall is an important feature of the present system. With self-propelled objects, this naturally raises the question of permanent fluxes in the system, a property forbidden in equilibrium wetting or adsorption. Therefore, we calculated the local particles flux **J(r)** (see Methods) predicted by our ABP model and show the result in Fig. 3C. We observe an upward flux at the attractive and aligning wall in the dilute regime, consistent

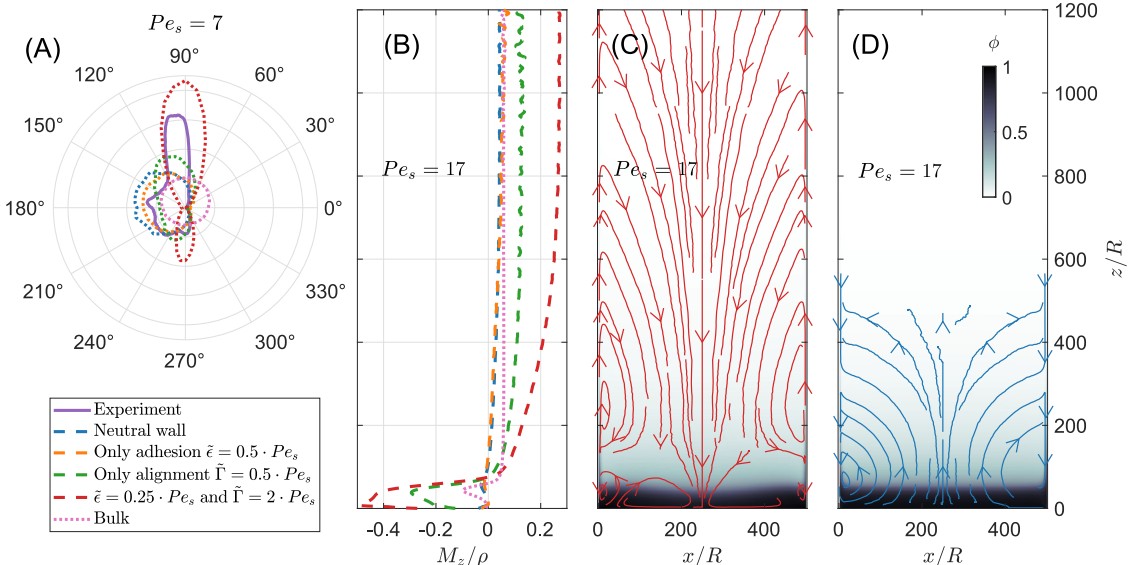

**Fig. 3 | Orientation distribution, polarization and fluxes. A** Orientation distribution of particles near the wall at activity $Pe_s = 7$. We consider in our ABP model (dotted lines) as well as in the experiment (purple solid line) the situation which corresponds to the left wall ($x = 0$) in Fig. 1E. Thus 0° corresponds to a particle oriented away from the wall and 90° is equivalent to a particle oriented against gravity. The numerical bulk distribution is also plotted in pink. **B** The numerical profiles of the $z$-component of the polarization (particle orientation) $M_z/\rho$ in the *adsorption layer* (dashed lines) and in the bulk (dotted line) at $Pe_s = 17$ for the four cases: a neutral wall (blue), an adhesive wall (orange), an aligning wall (green) and a wall with alignment plus attraction (red). Particles point upwards if $M_z > 0$ and downwards if $M_z < 0$. **C, D** Numerical streamlines of a velocity field $\mathbf{v} = \mathbf{J}/\rho$ together with the packing fraction field $\phi$ for a simultaneously attracting and aligning wall with $\tilde{\epsilon} = 0.25 Pe_s = 4.25$ and $\bar{\Gamma} = 2 Pe_s = 34$ **C** and for a neutral wall **D**.

with the enhanced upward polarization (Fig. 3A, B). Unlike in the (unconfined) bulk, the balance between gravity and upward propulsion is broken at the wall.

This fundamentally out-of-equilibrium response, which involves the presence of permanent fluxes in the steady-state, allows us to propose a rationale for the complex dependence of the rising height $\Delta H$ on aligning wall interactions (Fig. 2A). When the alignment comes with an additional *direct* wall adhesion, particles are trapped long enough at the wall for the extra upward polarization to induce an upward active flux that is larger than the sedimentation-canceling bulk reference. Accordingly, we expect a high rise of the particles at the wall, which is of purely dynamical origin.

Moreover, an aligning wall also enhances the downwards polarization present in the dense part of the bulk. Consistently, it causes a strong downward flux close to the wall, resulting in a well-pronounced vortex in the bulk. As pointed out, the existence of steady-state particle currents is a strong signature of the non-equilibrium nature of active adsorption[44]. Note that, experimentally, so far, we can only measure the downward flux in the dense region close to the wall (see Supp. Mat.). We do not have enough statistics to measure experimentally the upward flux.

Thus, the total currents are only accessible in our ABP model. For the neutral wall (Fig. 3D), we observe a small downward current close to the wall. Such current can be explained by the polarization decrease at the wall, breaking the nearby flux balance and generating a downward current. It should be noted that for a neutral wall these stationary downward currents along the walls together with a large clockwise/counter-clockwise rotating vortex in the right/left lower corner emerges also without particle-particle interactions in the ideal ABP gas and is a direct consequence of confinement[45].

As we just showed, dynamical aspects are the key to the out-of-equilibrium active phenomenon we report. So far, we have mostly discussed it in terms of single particle properties at the wall. However, when zooming in on this wall layer, we see it forms an assembly of 1D clusters, which we denote as *trains*, since they move collectively along the wall (see inset of Fig. 4 and movies 1, 2, 3 in Sup. Mat.). In a recent study of Janus particles orbiting along a circular post in quasi-2D[32],

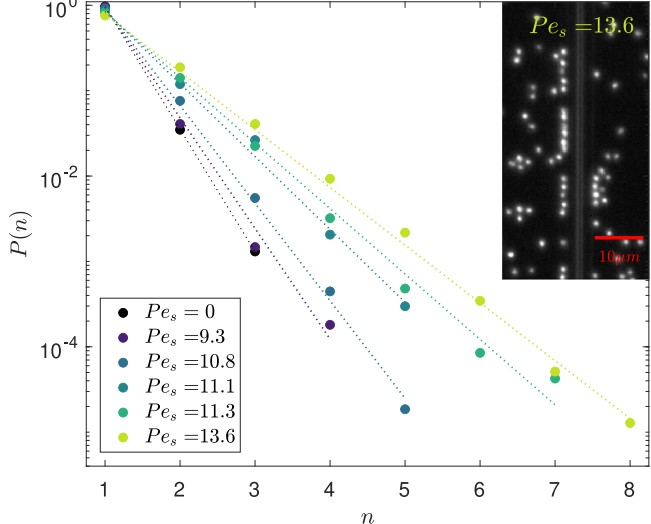

**Fig. 4 | Trains statistics.** Experimental probabilities of the train size in the adsorption layer $P(n)$ for different activities $Pe_s$. Two colloids form a pair if the distance between the centers is below 1.9 μm ≈ 2.4$R$. Dotted lines represent a geometric law fit to the data (See Methods). Inset: Snapshot of active Janus colloids sticking to the wall and forming trains on both sides of the wall.

similar train structures were observed resulting from collisions as well as hydrodynamic and osmotic interactions of 1D-moving swimmers. Although large trains tend to move slower than single particles at the wall, our trains are dynamical structures. They can merge or break apart. Particles leave the trains at their ends or are squeezed out into the second, more labile, layer. Whilst these structures catch the eye, we could not find any signature of physical significance, e.g. wall-induced phase separation. For instance, the density distribution at the wall is not bimodal (not shown). Instead, the presence of trains can be explained by pure randomness. We focus in Fig. 4 on the probability

distribution $P(n)$ of train size $n$ in the experiments. Note that the numerical model also exhibits such trains (see Supplementary Fig. 8). We observe that the distribution decays exponentially and thus is completely dominated by monomers. To describe such behaviour, we examine the possible origin of trains in our system, and the extent to which trains can appear randomly, without an underlying formation mechanism. To do so, we consider a simple site-adsorption model where the adsorption layer is considered as a 1D system with homogeneously distributed adsorption sites. These sites are randomly populated by adsorption-desorption exchanges with the nearby bulk reservoir (see Methods). The probability of having a train of size $n$ is very well described by such a random site-adsorption model, as shown in Fig. 4. The dotted lines correspond to geometric laws, where the average density is the only adjustable parameter. The values obtained fall within the range of experimentally measured densities (see Sup. Mat.) Overall, there is no indication of collective dynamics on the wall layer which justifies the single particle arguments used so far, although we cannot rule out more subtle collective effects.

## Discussion

To summarize, we study experimentally and theoretically the behaviour of an assembly of active colloidal particles in the presence of a vertical confining wall and gravity and explore the possibility of having an active capillary rise. Even in the absence of (motility-induced) phase separation, we observe a dynamic adsorption layer at the wall that rises with activity, and we demonstrate that this rise cannot be explained by a simple mapping to an effective attraction. We find that the interaction between the particles and the wall has a significant impact on this layer, but that the height of the layer cannot be simply understood in terms of detention time as in a gravity-free situation. In particular, we show that gravity is essential to generate a polarization in the bulk, that is, then enhanced by wall-alignment. This polarization, together with activity-dependent wall-adhesion, is most likely responsible for the persistent vertical pumping we observe in the system.

How does this persistent pumping fit into the broader context of active matter studies?

The exploration of mechanical aspects in active matter, such as the concept of pressure has triggered an abundant literature illustrating how active matter can depart from equilibrium systems[14,15]. These peculiarities of active systems become evident when they interact with walls or interfaces, at the core of striking features such as work extraction from a bath with ratchet-like rotors[2]. Following this appealing route, the extension of wetting phenomena to active matter was recently considered[26]. In MIPS systems, activity provides an effective attractive inter-particles interaction responsible for the phase separation into a dense active-liquid phase. It also provides an effective wall-particle attraction so that phase separated systems in contact with a solid wall display the global phenomenology obeyed by wettable walls in contact with a liquid: macroscopic ascending meniscus, capillary rise, etc.[26]. Also starting from phase-separated systems, but here made of classical polymer-polymer solutions, a rich panel of phenomena were evidenced when adding activity into one of the phases[22]. Indeed, activity promoted the transition from a highly wetting meniscus to a full-wetting configuration reminiscent of the Landau-Levich coating transition[46]−although this was not discussed along this line in[22]. On the contrary, the present system does not phase separate and was only shown to form clusters[28,47]. The dense phase is held together mostly because of gravity, and overall no macroscopic meniscus forms at the wall whatever the conditions explored. In that respect, the wall layer that forms with activity would be closer to the adsorption of supercritical fluids at solid surfaces[48]. At odds though with this equilibrium analogy, this adsorption layer is associated with strong dynamical effects with steady-state fluxes across the system.

Our results demonstrate that a vertical wall effectively harvests energy from the microscopic scale to produce a macroscopic work.

More generally, a side wall can act as a pump against a force parallel to it, generating a net steady-state flux in the system. These results pave the way for active microfluidic systems, where even a basic configuration involving walls and gravity could play a role analogous to a generator in an electric circuit.

## Methods

### Experimental set-up

Gold particles of radius $R = 0.8\,\mu m$ were grafted with octadecanethiol[49] and half-coated with Platinum to form Janus microswimmers when immersed in hydrogen peroxide ($H_2O_2$)[27,47]. Due to their high mass density $\mu \simeq 11\,g/cm^3$, the particles immediately sediment onto the flat bottom of the experimental cell, forming a bidimensional monolayer of sedimented active particles. A very low in-plane apparent gravity $\vec{g}^*$ is obtained by tilting the whole set-up with a small angle $\theta \approx 0.1°$ in the $z$ direction. An elongated borosilicate micropipette bent on the bottom of the observation cell and dipped into the 2D sediment acts as a lateral wall. We focus on the half-space on the right side of the wall. As the surface of the glass is charged, colloidal particles are repelled at short range and there is no solid contact or friction between particles and wall.

By tuning $H_2O_2$ concentration from $c_0 = 3.0 \times 10^{-4}$ v/v up to $5c_0$, it is possible to vary the activity of the colloid. In practice, for each experiment, we characterize this activity by measuring, from the sedimentation profile in the dilute regime, the sedimentation length $\lambda_0$ in the passive case and the sedimentation length $\lambda$ for each active case[28,29]. The $Pe_s$ is then determined by $Pe_s = \sqrt{\frac{8}{3}(\lambda/\lambda_0 - 1)}$ [50]. This definition as been shown to be equivalent to the definition of the Péclet number from swim velocity[4]. For each concentration in $H_2O_2$, 3000 images of 2048 pixel × 2048 pixel are recorded at 5 fps using a Basler camera (ac-A2040-90 μm) mounted on a Leica DMI 4000B microscope and a custom-made external dark field and a 20× fluotar objective. The pixel size is 0.273 μm. We track the positions of the center of the particles using Trackpy toolkit on Python[51]. We measure, within one pixel (30% of $R$) the most occupied position $x_m$ close to the glass wall. We set the origin of the $x$ axis ($x = 0$) at $0.5R$ before $x_m$. We define the *adsorption layer* as the layer between $x = 0$ and $x = L = 2R$. Note that the average equilibrium distance between particles is $2.4R$.

Our observable for the analysis are the density maps $\rho(x, z)$, that are obtained by averaging over time the number of particles for each pixel divided by its surface $7.45 \times 10^{-2}\,\mu m^2$. The density profiles of the bulk are computed as $\phi_{bulk}(z) = \frac{\pi R^2}{x_r - x_l} \int_{x_l}^{x_r} \rho(x, z)\,dx$ with $x_l = 80R$ corresponding to a value far enough from the wall, at least six times the average equilibrium distance between particles, and $x_r = 340R$, corresponding to the border of the image. The density profile in the adsorption layer is obtained via $\phi_{wall}(z) = \frac{\pi R^2}{L} \int_0^L \rho(x, z)\,dx$. From both density profiles, adsorption layer heights can be determined by measuring the difference in altitude using a 0.01 broad interval centered on $\phi = 0.08$.

We define the *trains* by considering the set of particles in the adsorption layer whose centers are separated by a distance of less than $2.4R$. The train size distribution was analyzed for trains at altitudes higher than, $z = 300\,\mu m$ corresponding to a position at which the bulk density is approximately 0.08 for all studied activities.

*Polarity* is measured on different experiments, on the same system, recorded in reflection with a Baumer HGX40c color camera, a 60x objective and a 1.6x zoom. The polarity of a particle is given by the shift between its position on the green channel and its position on the blue channel, corrected for chromatic aberration, as explained in Supplementary Methods.

### Random site-adsorption model

We look at the statistics of trains, that is, of segments of continuously populated sites. We note $p$ the probability that a site is occupied by a

particle and $q = 1 - p$ the probability that a site is empty. $p$ corresponds to the mean lineic fraction of particles along the wall. The probability that a random chosen site belongs to a train of size $n$ is proportional to $np^n(1-p)^2$. From which we derive the probability of having a train of size $n$ as $P(n) = (1-p)p^{n-1} = q(1-q)^{n-1}$ that abides a geometric law.

## Active Brownian Particle (ABP) model

We model the self-propelled particles as two-dimensional active Brownian particles swimming with a constant velocity $v_0$ in a container of size $L_x$ and $L_z$ along the $x$- and $z$-direction, respectively. The position $\mathbf{r}_i = (x_i, z_i)$ and the orientation $\mathbf{e}_i = (\cos\theta_i, \sin\theta_i)$ of the $i$-th particle evolve according to the overdamped Langevin equations:

$$\dot{\mathbf{r}}_i = v_0\mathbf{e}_i + \gamma_t^{-1}\mathbf{f}_i - v_g\mathbf{e}_z + \sqrt{2D_t}\,\boldsymbol{\eta}_i \tag{1}$$

$$\dot{\theta}_i = \gamma_r^{-1}t_i^{\text{wall}} + \sqrt{2D_r}\,\xi_i \tag{2}$$

for $i = 1, ..., N$. The Einstein relation for translation and rotation is $\gamma_t = k_B T_0/D_t$ and $\gamma_r = k_B T_0/D_r$, respectively, where $\gamma_t$ and $\gamma_r$ are the friction coefficients, $D_t$ and $D_r$ the diffusion constants and $k_B T_0$ the thermal energy. $\boldsymbol{\eta}_i$, $\xi_i$ are zero-mean unit-variance Gaussian white noises. For a spherical Brownian particle, we have $D_r = 3D_t/(2R)^2$. Due to the reduced gravity, particles sediment with velocity $v_g$ along the negative $z$-direction $-\mathbf{e}_z$. The force on $i$-th particle $\mathbf{f}_i$ consists of a part due to particle-particle, $\sum_{j\neq i}\mathbf{f}_{ij}$, and particle-wall interaction, $\mathbf{f}_i^{\text{wall}}$. The particles interact via a repulsive pair potential $V(r) = \frac{k}{2}(2R-r)^2$ if $r \leq 2R$, i.e., the inter-particle distance $r$ is smaller than the particle diameter $2R$, and $V(r) = 0$ otherwise[36]. The repulsion strength $k$ is chosen such that the particle overlap is 0.01 of the diameter $2R$ during a head on collision. The force on $i$-th particle due to $j$-th particle reads as $\mathbf{f}_{ij} = \mathbf{f}(\mathbf{r}_i - \mathbf{r}_j) = -\nabla_{\mathbf{r}_i}V(|\mathbf{r}_i - \mathbf{r}_j|)$. To account for a possible wall adhesion, we let the particles interact with walls via a Lennard-Jones potential (see Supplementary Materials for alternative choice of potential), which for the left wall at $x = 0$ reads as

$$V_{\text{left}}^{\text{wall}}(x) = 4\epsilon\left[\left(\frac{R}{x}\right)^{12} - \left(\frac{R}{x}\right)^6\right], \tag{3}$$

where $\epsilon$ controls the attraction to the wall, and similarly for the right, bottom, and top wall. For a neutral (purely repulsive) wall, we use a Lennard-Jones potential truncated at $x = 2^{1/6}R$ and set $\epsilon = k_B T_0/2$. In order to mimic a possible bipolar (or nematic) wall alignment, the particles experience a position-dependent torque, which for the left wall ($x = 0$) reads as

$$t_{\text{left}}^{\text{wall}} = \Gamma\sin(2\theta)\left(\frac{R}{x}\right)^3, \tag{4}$$

where $\Gamma$ is the strength of alignment. For $\Gamma > 0$ this torque orients the particles parallel to the wall. The form of $t^{\text{wall}}$ is motivated by hydrodynamic interaction of force dipole microswimmers with surfaces[34], as Janus colloids were experimentally characterized as pushers[40]. Four dimensionless numbers govern the system. A measure of activity is the normalized swim persistence length, the swim Péclet number $\text{Pe}_s = v_0/(RD_r)$. The range of activities considered here was $5 \leq \text{Pe}_s \leq 17$, which is below the critical point $\text{Pe}_{\text{crit}} \approx 26.7$[38]. Similarly, one can define a gravitational Péclet number as $\text{Pe}_g = v_g/(RD_r)$, which we kept fixed to the experimental value $\text{Pe}_g = 1$. Furthermore, we define a dimensionless adhesion and alignment strength as $\tilde{\epsilon} = \epsilon/k_B T_0$ and $\tilde{\Gamma} = \Gamma/k_B T_0$, respectively. Our simulation box was of size $L_x = 500R$ and $L_z = 1500R$ and contained $N = 14000$ particles. The simulation time was at least $750000/D_r$. For comparison, the observation time in the experiment was approximately $300/D_r$.

## Microscopic fields

The density field at position $\mathbf{r}$ is given by

$$\rho(\mathbf{r}) = \left\langle\sum_{i=1}^{N}\delta(\mathbf{r}-\mathbf{r}_i)\right\rangle, \tag{5}$$

where the angles denote a statistical average, $\delta$ the Dirac delta function, and $\mathbf{r}_i$ the position of $i$-th particle. The microscopic fields are coarse-grained using a Gaussian kernel instead of a delta function[52,53]. The local polarization is defined as

$$\mathbf{M}(\mathbf{r}) = \left\langle\sum_i\mathbf{e}_i\,\delta(\mathbf{r}-\mathbf{r}_i)\right\rangle, \tag{6}$$

where $\mathbf{e}_i$ is the orientation of $i$-th particle. The particle flux $\mathbf{J}(\mathbf{r})$ is obtained from the force density balance equation

$$\mathbf{J} = v_0\mathbf{M} + \gamma_t^{-1}\mathbf{F} - v_g\mathbf{e}_z\rho - D_t\nabla\rho, \tag{7}$$

where $\mathbf{F}$ denotes the internal force density field[54]. Each term of Eq. (7) can be measured easily in simulations.

## Data availability

Data that support the findings of this study are available from the corresponding authors upon request. Source data are provided in this paper.

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

## Acknowledgements

A.F.C. is supported by PhD scholarship from the doctoral school of Physics and Astrophysics, University of Lyon. C.C.B. and C.Y. acknowledge financial support from ANR-BACMAG. A.W. and H.R. acknowledge financial support from the German Research Foundation (DFG), project 580/15-1 and INST 256/539-1.

## Author contributions

A.F.C. and A.W. contributed equally to this article. H.R. and C.C.B. designed the original project. A.F.C. performed experiments and data analysis. A.W. performed numerical simulations and data analysis. M.L. performed the polarity analysis. C.Y. provided theoretical justification for the wall interaction used in simulations. A.W., C.Y., M.L., H.R., and C.C.B. guided the research. All authors interpreted the results and contributed to the manuscript.

## Funding

## Competing interests
