## [Peer Review File · Nature Communications]

REVIEWER COMMENTS

Reviewer #1 (Remarks to the Author):

In this article, Fins Carreira et al. demonstrate a wetting-like behaviour of gold-platinum Janus colloids in hydrogen peroxide. This article is a continuation of (a subset) of the authors' earlier articles. I think the results that the authors present are interesting, and the article is very well-written. However, the result that the presence of a wall breaks the equilibrium-like mapping is not very surprising (indeed, if anything, the equilibrium-like mapping is the surprise). As such, I think this article may not be of great interest to a broad audience and belongs to a more specialised journal.

I should qualify what I mean by the statement that the "presence of a wall breaks the equilibrium-like mapping is not very surprising". Effective equilibrium mappings in driven systems are incredibly delicate. Even the effective equilibrium mapping for sedimenting ABPs really only works when one only tracks the density field. As the authors themselves discuss, there is a mean polarisation which doesn't really have a passive analogue. It is by now also well-known that (non-sedimenting) active particles near a wall have a net polarisation (authors refer to some relevant articles here; among many others, see Duzgun and Selinger PRE 97, 032606), particle distribution near a boundary is higher than the bulk and is related to the curvature (again, the authors refer to many of the relevant articles; among others, see the works by Fily, Baskaran and Hagan; for instance Soft Matter 2014). The point is that it is by now well understood that active units behave differently in the bulk and near the boundary, and an equilibrium-mapping may not work both in the bulk and at the boundary. The present article is, of course, a further new and interesting example of active units behaving differently at boundaries.

I have several more minor comments and questions.

1. It is a pity that one of the most striking effects discussed by the authors -- circulating steady-state currents -- is not accessible experimentally. Could the authors find some (even indirect) signature of this in their experiments?

2. In Fig. 2A, the active plot has two distinct (but related) effects of activity -- just the bare activity and activity-induced adhesion. Can activity by itself (without creating adhesive effects which may be present) on ΔH simply mimic adhesion? What I mean is the following: is the effect of increasing activity at a neutral wall (with no adhesion or alignment) on ΔH exactly equivalent to increasing adhesion in a passive (but adhesive) system?

3. The data for detention time is somewhat inconclusive and doesn't really follow the trends of any of the numerical plots (to me, it doesn't seem to follow the adhesion-only curve; instead, it seems to go from a higher to a lower value and saturate). Also, in Fig 2B, the best fit line for the experimental point (which the authors did not plot) seems to have a significantly higher slope than all the numerical lines in that figure. Further, in Fig. 3A, the wall-polarisation is much more pronounced in the experiment than in the theoretical models. Given that the experimental and theoretical models seem to match only very qualitatively (and, in the one case, not even that), could the authors speculate on what the most crucial ingredient missing in their model is? Of course, I do understand that the ABP model doesn't aim at a systematic recapitulation of the experimental results but an impressionistic one.

4. Why do the authors choose an LJ potential for adhesion? I think the form of the adhesion force can be calculated for a diffusiophoretic swimmer (the authors only attempt to justify that the coefficient of the LJ potential is proportional to Pe in the supplement; not why an LJ potential is a good choice). Can the authors try to use a more realistic adhesion interaction? Could this be the cause of the discrepancy between the experimental and the numerical results (and how much do the numerical results depend on the specific form of the adhesive interaction)?

Reviewer #2 (Remarks to the Author):

The authors have performed a very interesting work merging motility induced phase separation and active wetting. The main take away is that a combination of self propulsion through persistence of motion, and alignment interaction, can orient Janus particles in the vertical direction of a wall to escape a sediment. The work is presented in an intuitive way and very well supported by both experimental data and numerics.

I have a main question about a point that I didn't get from the manuscript. The authors present a certain interaction between attractive interactions to the wall, alignment interactions, also with the wall, and persistence of motion. As explained in page 3, in the absence of attractive interactions with the wall, particles will stick to it only through persistence of motion, in a process similar to what induces MIPS, and particles will escape the wall when their direction points away from it through diffusion. An alignment interaction can make escape easier, as it will push the angle closer to the escape angle, which without interactions is simply $\theta = 0$ (I'm interpreting $\theta = 0$ to be pointing up, $\theta > 0$ pointing away from the wall). Attractive interactions will then naturally make the escape angle some value $\theta > 0$

A combination of alignment and attraction will then stabilise this climbing, as the authors say the angle will be pushed towards $\theta=0$, which is now below the escape angle. It was not clear to me where this alignment interaction, which seems important to observe this wetting phenomena, comes from. I had the idea that friction with walls would put a torque on self propelled particles that would make θ approach $-\pi/2$. As the authors mention in the Methods section, dipole forces can exhibit certain active anchoring properties that can align this dipoles with interfaces or walls. If this is the process that drives alignment, it should be discussed more, since it means alignment comes from a hydrodynamic effect. Initially I thought there was contact between particles and wall, given that the attraction between them is an electrostatic property and not a hydrodynamic one. I might be wrong here, and I suppose I am given their experimental results, but I would like this clarified.

The detention time the authors report seems to be incompatible with their model. Both adhesion and alignment create the behaviour one would expect, that is, a larger detention time as a function of activity. Given that this combination of adhesion and alignment seems to be the most relevant mechanism in this wetting process, it's a bit worrying that detention time actually decreases with activity in the experiments. The authors make a short comment about extending the model with hydrodynamic interactions, but I believe this contradiction should be commented in more detail, given that the trend predicted is the complete opposite of the trend observed.

Finally, their model for "trains" is very different from the general model based on ABPs. Does the ABP model show anything resembling this climbing clusters? I suppose if all particles have the same velocity, once they're climbing one wouldn't expect them to slow down or speed up enough to cluster, while experiments have fluctuations in the velocity that may facilitate this clustering, together with other attractive interactions between them. At the same time, the model has attractive interactions so I wouldn't be surprised if particles could cluster. A brief comment on the presence of clusters in the numerical model would be appreciated.

I believe the community will find this work very relevant and will inspire related studies. As such, pending answers to these questions, I think it deserves publication and Nature Communications is the right journal for it.

REPLY TO REVIEWER 1

In this article, Fins Carreira et al. demonstrate a wetting-like behaviour of gold-platinum Janus colloids in hydrogen peroxide. This article is a continuation of (a subset) of the authors' earlier articles.

We thank the Referee for their careful read of our manuscript in the context of the literature in general and our previous works in particular. However, we would like to stress that our manuscript is neither a continuation of earlier work nor incremental research: the central question that we address in this manuscript has never been addressed before experimentally: does polar active matter show a capillary rise? Of course we use our experimental workhorse, which includes specific active colloids and a specific setup to exert a homogeneous gravitational force, but in order to address the central question we had to develop new analysis methods and computational models. As described below, we have extensively amended the text to make this message clearer.

I think the results that the authors present are interesting, and the article is very well-written. However, the result that the presence of a wall breaks the equilibrium-like mapping is not very surprising (indeed, if anything, the equilibrium-like mapping is the surprise).

We thank the Referee for their assessment that our results are interesting and our manuscript is well written. However the remark of the Referee made us painfully aware that the previous version of the manuscript was, unbeknownst to us, misleading about the main message we wanted to convey. Indeed, our main message is NOT “*that presence of a wall breaks the equilibrium-like mapping*”. On the contrary, it is well known by now that active matter cannot be described by or be mapped onto equilibrium systems. The urgent question is now to reveal and to understand the non-equilibrium characteristics of active matter quantitatively, to which we contribute in our manuscript. We tried to figure out how we could avoid the impression that our aim could be to demonstrate that some ‘equilibrium mapping’ fails, and instead stressed our main message: does polar active matter show a capillary rise?.

In the introduction, the following sentences were probably misleading:

“To date, such active energy harvesting did not show up in the ubiquitous configuration of self-propelled particles exposed to a constant and uniform force, e.g. gravity. There, indeed, active sedimentation properties were shown to be essentially that of a hotter suspension. In this paper, we explore how this picture may change drastically when bringing a lateral confining wall in contact with sedimenting polar active particles, a configuration which may fall within active wetting phenomena. ”

We changed these sentences into

“To date, such active energy harvesting **has not been exploited** in the ubiquitous configuration of self-propelled particles exposed to a constant and uniform force, e.g. gravity. There, **so far**, active sedimentation properties were shown to be **roughly** that of a hotter suspension, **although finer examination reveals intrinsic out-of-equilibrium characteristics**. In this paper **we address the question: Does polar active matter exhibit capillary rise? To investigate this question we introduce** a lateral wall in contact with sedimenting polar active particles, a configuration which may fall within active wetting phenomena.” (emphasis added on the differences)

We also clarified the end of the introduction as

“Here, we present experimental observations of the behavior of a non phase separating assembly of active colloids that are sedimenting in a gravitational field in the presence of a lateral wall **and explore the question of active capillary rise for polar active matter**. We provide the evidence **that even without phase separation polar active matter can display** an unexpected and dynamic adsorption layer at the wall that **rises against gravity and** has no analogue in passive systems.” (emphasis added on the differences)

Throughout the previous version of the manuscript, we were parameterizing the activity of the system using a quantity that we named "effective temperature" T_{eff} , which is essentially the square of the Péclet number. Since "temperature" is commonly associated with "equilibrium" we learned from the Referees misunderstanding that we should avoid this name - and replaced T_{eff} by the Péclet number everywhere. With this modification it should be clear that we do not have any 'equilibrium mapping' in mind.

Confusion may also have arisen from

“This is what is expected for activity resulting in a simply hotter system with a higher sedimentation length (Fig.1 C-D). Strikingly, this hot-colloids mapping breaks down in the vicinity of the wall, where we evidence an upturn of the iso-densities that rise with increasing activity.”

that we changed to

“This is what is expected **in the simplistic picture where activity results in a** hotter system with a higher sedimentation length (Fig.1 C-D). **Nevertheless**, this **simplistic** hot-colloids mapping **strikingly** breaks down in the vicinity of the wall, where we evidence an upturn of the iso-densities that rise with increasing activity.”

At the end of the first description of the observed phenomenology, we highlighted that our goal was not ‘equilibrium mapping’ by adding

“Therefore the layer we observe is not wetting layer per say (implying a gas-liquid phase separation), but an adsorption layer. In the following we investigate the physical origin of this adsorption, how it interacts with gravity and how this phenomenology is specific to polar active matter and has no equilibrium analogue.”

Finally, at the beginning of the discussion, we recall that we

“study experimentally and theoretically the behavior of an assembly of active colloidal particles in the presence of a vertical confining wall and gravity **and explore the possibility of having an active capillary rise. Even in the absence of (motility induced) phase separation,** we observe a dynamic adsorption layer at the wall that rises with activity, and we **demonstrate that this rise cannot be explained by a simple mapping to an effective attraction.**”

We hope that these changes of phrasing make clear what is and is not the message of our paper and raise its significance to the eyes of the Referee.

As such, I think this article may not be of great interest to a broad audience and belongs to a more specialised journal.

We hope that provided the above changes, and the understanding that the central question that we address in this manuscript is ‘does polar active matter show a capillary rise?’, the Referee will reconsider their assessment. Indeed, this question has never been addressed before experimentally. Our manuscript answers to this question in the affirmative, quantify this capillary rise and reveal the mechanism leading to it. This is the first paper reporting **experimental** evidence for a capillary rise in **polar** active matter.

Recently wetting phenomena of **any** active matter systems has been of great interest for a broad audience, as can be assessed for instance from the venues that published the few works that have addressed this issue up to now. Capillary rise of polar active matter has been predicted **theoretically** only recently by two of us (Phys. Rev. Lett. 124, 048001 (2020)) and a related wetting phenomenon has only recently been reported for an active **nematics** by Adkin et al. (Science 377, 768 (2022)), which rises at a wall due to a mechanism that is completely different from what happens in **polar** active matter, as we describe both numerically and experimentally in our manuscript.

We hope that the general interest of capillary rise in the context of active matter has not been quenched by the two aforementioned papers. We also hope that the Referee will now recognize that the novelty of our experimental and numerical results in the vast class of polar active matter makes our manuscript suitable for the broad readership of Nature Communications.

I should qualify what I mean by the statement that the "presence of a wall breaks the equilibrium-like mapping is not very surprising". Effective equilibrium mappings in driven systems are incredibly delicate. Even the effective equilibrium mapping for sedimenting ABPs really only works when one only tracks the density field. As the authors themselves discuss, there is a mean polarisation which doesn't really have a passive analogue. It is by now also well-known that (non-sedimenting) active particles near a wall have a net polarisation (authors refer to some relevant articles here; among many others, see Duzgun and Selinger PRE 97, 032606), particle distribution near a boundary is higher than the bulk and is related to the curvature (again, the authors refer to many of the relevant articles; among others, see the works by Fily, Baskaran and Hagan; for instance Soft Matter 2014). The point is that it is by now well understood that active units behave differently in the bulk and near the boundary, and an equilibrium-mapping may not work both in the bulk and at the boundary. The present article is, of course, a further new and interesting example of active units behaving differently at boundaries.

We agree with the Referee that mapping the behaviour of active matter to equilibrium is at best simplistic. Some of us indeed helped established the 'hot colloid' approximation for sedimenting ABPs 13 years ago (see PRL 105 0909.4193 2010), but were also among the first to measure polarization effects that are specific to polar active matter in this very case (see NJP 20, 115001, 2018). As stressed by the Referee, we are also aware of the work of others in gravity-free situations that showed that active matter generally behave differently in the bulk and near the boundary. We have complemented our existing discussion of these points by the two references suggested by the referee. With the changes listed above, we hope to have made clear that the goal of this review of the literature was not to claim 'equilibrium mapping', but to provide the context of the question we address in this manuscript: 'does polar active matter show a capillary rise?' By answering this specific question we want to understand in general **how** active matter behaves at walls and what are the physical mechanisms for what it does. As our manuscript demonstrates, adding an external field – here gravity – unravels new behaviours specific to polar active matter. We hope that the Referee will agree that this is more accurate and of more interest than the general claim that active matter behaves differently from an equilibrium system especially at boundaries.

1. It is a pity that one of the most striking effects discussed by the authors – circulating steady-state currents – is not accessible experimentally. Could the authors find some (even indirect) signature of this in their experiments?

We agree with the Referee; it is indeed unfortunate that we cannot access these fluxes experimentally. To obtain the circulating steady-state currents numerically and reduce noise, the

data are averaged over a time of $10^6\tau_r$, which, unfortunately, is not achievable in the experiments. We have made our best efforts to explore experimental possibilities to extract signals from the noise. In the experiments, we were able to measure the vertical flow in the dense part in the adhesion layer and extract signals from the noise in the dense part of the system. In this region, we observe a negative flux, corresponding to downward flow, measured numerically (see new Supplementary Figure 7). These data are now included in the supplementary materials.

2. In Fig. 2A, the active plot has two distinct (but related) effects of activity – just the bare activity and activity-induced adhesion. Can activity by itself (without creating adhesive effects which may be present) on Delta H simply mimic adhesion? What I mean is the following: is the effect of increasing activity at a neutral wall (with no adhesion or alignment) on Delta H exactly equivalent to increasing adhesion in a passive (but adhesive) system?

Indeed activity by itself, without creating adhesive effects, can mimic adhesion on Delta H. This is something that we could observe in Fig 2A (previously Fig. 2B), where the blue squares correspond to the neutral wall, representing the results for pure activity and no adhesion. We do observe that pure activity mimics adhesion on ΔH , but the effect is lower than when there is a combination of activity and adhesion (orange circles), and even less pronounced than the combination of activity, adhesion, and alignment (red triangles). We have modified the text to make this clearer and have added the sentences

“We observe in this case that pure activity, on a neutral wall, generates an adhesion layer with ΔH increasing with Pe_s . Activity has thus an effect that mimics adhesion.”

However, in order not to muddy our message with tentative ‘equilibrium mapping’, we have not shown our attempts to find the equivalent direct adhesion to a given activity.

3. The data for detention time is somewhat inconclusive and doesn’t really follow the trends of any of the numerical plots (to me, it doesn’t seem to follow the adhesion-only curve; instead, it seems to go from a higher to a lower value and saturate). Also, in Fig 2B, the best fit line for the experimental point (which the authors did not plot) seems to have a significantly higher slope than all the numerical lines in that figure. Further, in Fig. 3A, the wall-polarisation is much more pronounced in the experiment than in the theoretical models. Given that the experimental and theoretical models seem to match only very qualitatively (and, in the one case, not even that), could the authors speculate on what the most crucial ingredient missing in their model is? Of course, I do understand that the ABP model doesn’t aim at a systematic recapitulation of the experimental results but an impressionistic one.

This comment, that was common to both Referees, greatly helped us discard a false equivalence we originally had in mind. Indeed, we now understand that the adhesion layer height is not described solely by detention time. Let us explain how we got to this conclusion.

In the previous version of the manuscript, well-aware of the limitations of the ABP description of our experimental system, we had refrained from fine tuning our adhesion and alignment parameters towards a ‘fit’ of the data. However, following the Referee’s very relevant comment, we have explored the adhesion and alignment parameters in order to find effective values evidencing behavior closer to the experimental one. For that we did a numerical mapping of $\tau_{detention}$ and ΔH on the $(\tilde{\epsilon}, \tilde{\Gamma})$ plane for a fixed $Pe_s = 13$ and compared it to the experimental measurements (see new Supplementary Figure 6). This led us to consider new values of $\tilde{\epsilon} = 0.25Pe_s$ and $\tilde{\Gamma} = 2Pe_s$. We have modified accordingly all the figures in the article considering those parameters. With those parameters, the detention time tends to saturate with Pe_s , and the numerical wall polarisation is much higher. The fact that the match with the experimental $\tau_{detention}$ is not perfect is not unexpected as many physical parameters are still missing in the simulations. Indeed, simple ABPs do not capture the complete dynamical behaviour of active colloids. Another step would be to refine the physical description to put in the simulation to have a proper description of the problem, but this is beyond the scope of this paper. This is discussed in the new section ‘Péclet number dependency’.

However - beyond the discussion of the relevance of the simulations to describe our experimental system - we noticed that there is no direct correspondence between $\tau_{detention}$ and ΔH , as strikingly illustrated in (Supplementary Figure 6). In particular, strong alignment and relatively weak attraction leads to large ΔH but detention times that are as short as with a neutral wall. Our new parameters to combine attraction and alignment in simulations fall in this category, as does our experimental system. In the new version of the manuscript, we highlight this decoupling and attribute it to the steady state flux. In particular, we have grouped together the graphs $\Delta H(Pe_s)$ and $\tau_{detention}(Pe_s)$ into a new Figure 2, specifically described in a new section called ‘Péclet number dependency’. At the end of this section we have added the following paragraph:

“Furthermore with this choice of parameters, although the detention time is similar to its values in the neutral wall case, the adhesion height is much larger. This is inconsistent with the simple picture that $\tau_{detention}$ drives ΔH . Indeed in the following, we shall see that part of the adhesion layer height is due to the singular influence that a wall parallel to gravity exerts on polarity and fluxes.”

In the discussion, we now state

“We find that the interaction between the particles and the wall has a significant impact on this layer, but that the height of the layer cannot be simply understood in terms of detention time as in a gravity-free situation. In particular, we show that

gravity is essential to generate a polarization in the bulk, that is then enhanced by wall-alignment.”

4. Why do the authors choose an LJ potential for adhesion? I think the form of the adhesion force can be calculated for a diffusiophoretic swimmer (the authors only attempt to justify that the coefficient of the LJ potential is proportional to Pe in the supplement; not why an LJ potential is a good choice). Can the authors try to use a more realistic adhesion interaction? Could this be the cause of the discrepancy between the experimental and the numerical results (and how much do the numerical results depend on the specific form of the adhesive interaction)?

The modelling of self-phoretic swimmers is actually a complex problem (Eur. Phys. J. Special Topics 225, 1843, 2016; The Journal of Chemical Physics, 4, 044902, 2019; The Journal of Chemical Physics, 4, 044901, 2019), where even the mere sign of interactions –attractive vs repulsive– and the presence or not of alignment depends on tiny physicochemical details of the particles that we cannot measure directly. Instead of this first-principle approach, we chose to use a simple potential that allows us to understand the basic ingredients that rule the experimental system behaviour. In addition to the LJ potential we also checked another standard potential used in computational models of active colloids, i.e. the Yukawa potential, and did not find significant differences in the observables that we measured (we show these results now in the Supplementary Material, in particular new Supplementary Figure 5). This is actually expected since the adhesive particle-wall interactions are very short-ranged (measured in terms of the distance from the "surface" of the nearly hard-core particles) such that the details of the mathematical model describing it do not matter. Actually a linear approximation in the interaction region (between the particle "surface" and the small adhesion interaction range) is sufficient, and for this one does not need to calculate the exact diffusiophoretic force.

We hope that the Referee will find that our revision in response to their concerns has improved our manuscript and that the revised manuscripts is now suitable for publication in Nature Communications.

REPLY TO REVIEWER 2

The authors have performed a very interesting work merging motility induced phase separation and active wetting. The main take away is that a combination of self propulsion through persistence of motion, and alignment interaction, can orient Janus particles in the vertical direction of a wall to escape a sediment. The work is presented in an intuitive way and very well-supported by both experimental data and numerics.

We thank the Referee for their careful reading of our manuscript and their kind assessment of our work.

I have a main question about a point that I didn't get from the manuscript. The authors present a certain interaction between attractive interactions to the wall, alignment interactions, also with the wall, and persistence of motion. As explained in page 3, in the absence of attractive interactions with the wall, particles will stick to it only through persistence of motion, in a process similar to what induces MIPS, and particles will escape the wall when their direction points away from it through diffusion. An alignment interaction can make escape easier, as it will push the angle closer to the escape angle, which without interactions is simply $\theta = 0$ (I'm interpreting $\theta = 0$ to be pointing up, $\theta > 0$ pointing away from the wall). Attractive interactions will then naturally make the escape angle some value $\theta > 0$. A combination of alignment and attraction will then stabilize this climbing, as the authors say the angle will be pushed towards $\theta = 0$, which is now below the escape angle. It was not clear to me where this alignment interaction, which seems important to observe this wetting phenomena, comes from. I had the idea that friction with walls would put a torque on self-propelled particles that would make θ approach $-\pi/2$. As the authors mention in the Methods section, dipole forces can exhibit certain active anchoring properties that can align these dipoles with interfaces or walls. If this is the process that drives alignment, it should be discussed more, since it means alignment comes from a hydrodynamic effect. Initially, I thought there was contact between particles and wall, given that the attraction between them is an electrostatic property and not a hydrodynamic one. I might be wrong here, and I suppose I am given their experimental results, but I would like this clarified.

The origin of the alignment interaction is an important element, and we apologize for not being clearer. The alignment arises indeed from hydrodynamic interactions of Janus colloids with the surfaces. Experimentally, Janus colloids were characterized from a hydrodynamic point of view as pushers. Aligning torque with surfaces for force dipole microswimmers has

been previously described in the literature. In our simulations, we have decided to incorporate the corresponding hydrodynamic aligning torque for pushers. Although there is no direct hydrodynamics in our simulation, we have used this effective hydrodynamic torque to account for the experimental hydrodynamic effects responsible for the alignment at the wall. We have amended the text and discussed this in the main part to avoid any confusion with direct contact between particles and walls. We have written:

“as well as a wall bipolar (also called nematic) alignment parallel to the wall **mimicking hydrodynamic torque**”

We also added to the methods

“As the surface of the glass is charged, colloidal particles are repelled at short range and there is no solid contact or friction between particles and wall.”

We hope this is clearer now.

The detention time the authors report seems to be incompatible with their model. Both adhesion and alignment create the behaviour one would expect, that is, a larger detention time as a function of activity. Given that this combination of adhesion and alignment seems to be the most relevant mechanism in this wetting process, it’s a bit worrying that detention time actually decreases with activity in the experiments. The authors make a short comment about extending the model with hydrodynamic interactions, but I believe this contradiction should be commented in more detail, given that the trend predicted is the complete opposite of the trend observed.

This comment, that was common to both Referees, greatly helped us discard a false equivalence we originally had in mind. Indeed, we now understand that the adhesion layer height is not described solely by detention time. Let us explain how we got to this conclusion.

In the previous version of the manuscript, well-aware of the limitations of the ABP description of our experimental system, we had refrained from fine tuning our adhesion and alignment parameters towards a ‘fit’ of the data. However, following the Referee’s very relevant comment, we have explored the adhesion and alignment parameters in order to find effective values evidencing behavior closer to the experimental one. For that we did a numerical mapping of $\tau_{detention}$ and ΔH on the $(\tilde{\epsilon}, \tilde{\Gamma})$ plane for a fixed $Pe_s = 13$ and compared it to the experimental measurements (see new Supplementary Figure 6). This led us to consider new values of $\tilde{\epsilon} = 0.25Pe_s$ and $\tilde{\Gamma} = 2Pe_s$. We have modified accordingly all the figures in the article considering those parameters. With those parameters, the detention time tends to saturate with Pe_s , and the numerical wall polarisation is much higher. The fact that the match with the experimental $\tau_{detention}$ is not perfect is not unexpected as many physical parameters are still missing in the simulations. Indeed, simple ABPs do not capture the complete dynamical

behaviour of active colloids. Another step would be to refine the physical description to put in the simulation to have a proper description of the problem, but this is beyond the scope of this paper. This is discussed in the new section ‘Péclet number dependency’.

However - beyond the discussion of the relevance of the simulations to describe our experimental system - we noticed that there is no direct correspondence between $\tau_{detention}$ and ΔH , as strikingly illustrated in (Supplementary Figure 6). In particular, strong alignment and relatively weak attraction leads to large ΔH but detention times that are as short as with a neutral wall. Our new parameters to combine attraction and alignment in simulations fall in this category, as does our experimental system. In the new version of the manuscript, we highlight this decoupling and attribute it to the steady state flux. In particular, we have grouped together the graphs $\Delta H(\text{Pe}_s)$ and $\tau_{detention}(\text{Pe}_s)$ into a new Figure 2, specifically described in a new section called ‘Péclet number dependency’. At the end of this section we have added the following paragraph:

“Furthermore with this choice of parameters, although the detention time is similar to its values in the neutral wall case, the adhesion height is much larger. This is inconsistent with the simple picture that $\tau_{detention}$ drives ΔH . Indeed in the following, we shall see that part of the adhesion layer height is due to the singular influence that a wall parallel to gravity exerts on polarity and fluxes.”

In the discussion, we now state

“We find that the interaction between the particles and the wall has a significant impact on this layer, but that the height of the layer cannot be simply understood in terms of detention time as in a gravity-free situation. In particular, we show that gravity is essential to generate a polarization in the bulk, that is then enhanced by wall-alignment.”

Finally, their model for "trains" is very different from the general model based on ABPs. Does the ABP model show anything resembling this climbing clusters? I suppose if all particles have the same velocity, once they're climbing one wouldn't expect them to slow down or speed up enough to cluster, while experiments have fluctuations in the velocity that may facilitate this clustering, together with other attractive interactions between them. At the same time, the model has attractive interactions so I wouldn't be surprised if particles could cluster. A brief comment on the presence of clusters in the numerical model would be appreciated.

The numerical ABP model also exhibits trains, and we have added a comment on that in the main text: *Note that the numerical model also exhibits such trains (see new Supplementary Figure 8)*. Additionally, we have created movies from the simulations for different parameters

and included them in the Supplementary Materials. This allows for easier comparison with the experiments. Thank you for highlighting this point.

We hope that the Referee will find that our revision in response to their concerns has improved our manuscript and that the revised manuscripts is now suitable for publication in Nature Communications.

REVIEWER COMMENTS

Reviewer #1 (Remarks to the Author):

I thank the authors for their revisions, which I think have improved the manuscript. I must also make a clarifying remark: When I said that the "article is a continuation...", I didn't mean it as a critique at all. It is important to understand multiple aspects of an experimental system.

The authors have answered my specific questions.

Any opinion about whether or not a manuscript is likely to be of popular interest is subjective (and very often turns out to be wrong). Since such judgements are likely to reveal more about the tastes and the interests of the reviewer, I guess the authors are correct that the only reasonable way to gauge whether something is of popular interest is to check how many other people are working on similar things. By that measure, I guess, active wetting is of broad enough popular interest to warrant publication of this article in this journal.

Reviewer #2 (Remarks to the Author):

The authors have satisfactorily answered most of my questions on my previous report.

I believe the results are interesting enough for Nature Communications given this can be motivate studies on energy extractio of active systems, a much more general field that has gained some traction in recent years.

My main two issues, however, still stand. The detention time seems to experimentally decrease with Peclet number, something the model only does clearly when only alignment is considered, and to a lesser extent with a neutral wall (Fig 2.).

It seems the chosen 'fit to data' was done in terms of the layer height ΔH , although it is not a good fit for the detention time. I understand the authors say the fit is not perfect, but it's not just this, it's that the trend is the complete opposite. If I take the image the authors now give, which is that detention time is just not a good descriptor of this phenomenon, then a comment on how to potentially extend the model would be appreciated. Indeed if this observable that they report is not properly captured, there should be at least a hint as to what is missing from the model. In their

response the authors say that they could refine the physical description but they don't say how. Just to be clear I don't think it is at this point of this work worth it to completely change the model, since it does capture some important aspects of the experimental system, but they should clarify what they believe is missing, even if left to future work.

Finally, my other comment was regarding the train structures they observe in both in the experiments and simulations. I think my comment still stands. It seems pretty clear in the videos of the simulations that these trains tend to be nonmotile, but jammed clusters of particles with their ends pointing into the cluster, that dissolve very quickly as their inner particles are pushed into the bulk. This would be consistent with a simple one dimensional MIPS happening at the walls. The experimental videos show the same thing happening, except some of these trains seem to actually be coherently motile, which would be at least compatible with particles with different velocities. Maybe I'm overthinking these structures, but are these anything other than 1D jamming on the wall?

In conclusion, pending these two short questions, I think the manuscript is worth publishing in Nature Communications, and will likely have an impact on the more general community.

REPLY TO REVIEWER 1

I thank the authors for their revisions, which I think have improved the manuscript. I must also make a clarifying remark: When I said that the "article is a continuation...", I didn't mean it as a critique at all. It is important to understand multiple aspects of an experimental system. The authors have answered my specific questions. Any opinion about whether or not a manuscript is likely to be of popular interest is subjective (and very often turns out to be wrong). Since such judgements are likely to reveal more about the tastes and the interests of the reviewer, I guess the authors are correct that the only reasonable way to gauge whether something is of popular interest is to check how many other people are working on similar things. By that measure, I guess, active wetting is of broad enough popular interest to warrant publication of this article in this journal.

We thank the Referee for their positive feedback.

REPLY TO REVIEWER 2

The authors have satisfactorily answered most of my questions on my previous report.

I believe the results are interesting enough for Nature Communications given this can motivate studies on energy extraction of active systems, a much more general field that has gained some traction in recent years.

We thank the Referee for their positive feedback and for their assessment that our results are interesting and can motivate studies on energy extraction of active systems.

The detention time seems to experimentally decrease with Peclet number, something the model only does clearly when only alignment is considered, and to a lesser extent with a neutral wall (Fig 2.).

It seems the chosen “fit to data” was done in terms of the layer height ΔH , although it is not a good fit for the detention time. I understand the authors say the fit is not perfect, but it’s not just this, it’s that the trend is the complete opposite.

Following both Referees’ previous comments, we had explored the adhesion and alignment parameters to identify effective values that demonstrate behavior closer to the experimental results. To achieve this we had conducted a numerical mapping of $\tau_{detention}$ and ΔH on the $(\tilde{\epsilon}, \tilde{\Gamma})$ plane (Figure S6 of Supplementary Materials). This study was performed only for a fixed $Pe_s = 13$ and we did not investigate the global influence of Pe_s . Due to the imperfections of the model, which we will discuss below, seeking a perfect match would be hardly reasonable. Nevertheless, when transitioning from $\tilde{\epsilon} = 0.5Pe_s$ and $\tilde{\Gamma} = 0.5Pe_s$ to $\tilde{\epsilon} = 0.25Pe_s$ and $\tilde{\Gamma} = 2Pe_s$ we observe on the figure below two main points. First, there was only a very slight change on the ΔH with these parameters sets. Second, there was an impact on the evolution of the detention time with Pe_s , it is now much flatter and closer to the experiments than before. Since the purely aligning wall shows a decreasing trend of $\tau_{detention}$ with Pe_s , we interpolate that even stronger alignment coupled

with even weaker adhesion would at some point lead to $\tau_{detention}$ decreasing with Pe_s as in the experiments. However, we chose not to explore higher parameters for the effective aligning torque, as they would correspond to effective parameters beyond a range comparable to the underlying physical phenomena. We preferred to be conservative and cautious in considering those values since the primary goal of this “fit” was mostly to provide a qualitative description of the picture.

More importantly, what this exploration of the parameter space taught us is that the detention time is not the sole driver of ΔH , there is also a strong influence of the polarization at the wall that depends on the particle-wall interactions.

If I take the image the authors now give, which is that detention time is just not a good descriptor of this phenomenon, then a comment on how to potentially extend the model would be appreciated. Indeed if this observable that they report is not properly captured, there should be at least a hint as to what is missing from the model. In their response the authors say that they could refine the physical description but they don't say how. Just to be clear I don't think it is at this point of this work worth it to completely change the model, since it does capture some important aspects of the experimental system, but they should clarify what they believe is missing, even if left to future work.

Indeed, this numerical ABP model is not perfect for precisely describing the observed phenomena. A more realistic model of an active colloid should take phoretic and hydrodynamic fields into account. A wall distorts both the chemical and the flow field around the particle as compared to those produced by a particle in bulk. The fields are coupled to each other; on the one hand, the chemical species around the particles are advected by the flow, and on the other hand, the slip velocity on the particle surface is determined by the gradient of a chemical field. The distorted fields act back on the colloid and lead to an additional translation and rotation as compared to the motion of a particle in bulk. Wall induced hydrophoretic interactions lead to a rich dynamics of a single particle, like attraction, repulsion or reorientation. The results are very sensitive to the model details,

Figure 1: Top: values of the adsorption layer height ΔH as a function of the activity Pe_s . Bottom: detention time of colloids without a neighbour in the *adsorption layer* versus the activity Pe_s . Data from experiment (purple dots) and from simulations, for a wall with both alignment and adhesion (red triangles) with $\tilde{\epsilon} = 0.25Pe_s$ and $\tilde{\Gamma} = 2Pe_s$ and (purple triangles) with $\tilde{\epsilon} = 0.5Pe_s$ and $\tilde{\Gamma} = 0.5Pe_s$.

however, several studies found a greatly enhanced swimming velocity parallel to the wall, which could be a possible mechanism leading to a high wetting height and at the same time to small detention times. Nevertheless, the incorporation of hydro-phoretic effects into a simulation is a rather involved task, beyond the scope of this paper.

We also clarified this by changing

“Achieving a more accurate agreement would require to refine the ABP model, particularly by including hydrodynamic and diffusiophoretic interactions.”

into

“Achieving a more accurate agreement would require to refine the ABP model, particularly hydrodynamic **interactions with the wall are missing. Also, the chemical gradient around the particle is modified by the presence of the wall, which should influence the velocity of the particle as well. A full description should involve taking into account both those hydrodynamics and diffusiophoretic effects and their coupling.**”

My other comment was regarding the train structures they observe in both in the experiments and simulations. I think my comment still stands. It seems pretty clear in the videos of the simulations that these trains tend to be nonmotile, but jammed clusters of particles with their ends pointing into the cluster, that dissolve very quickly as their inner particles are pushed into the bulk. This would be consistent with a simple one dimensional MIPS happening at the walls. The experimental videos show the same thing happening, except some of these trains seem to actually be coherently motile, which would be at least compatible with particles with different velocities. Maybe I’m overthinking these structures, but are these anything other than 1D jamming on the wall?

We deeply empathise with the Referee, as we have also spent a lot of time overthinking these eye-catching structures. We have also attempted to discern

the origin and significance of these structures in the light of the physics of active matter. For instance, we have examined the velocity of the trains as a function of their mean size, but did not observe anything noteworthy. We also searched for a bimodal density distribution along the wall, which could have been a signature of a 1D MIPS, but again, we did not observe this. Unfortunately, likely due to the exchanges with the bulk, we have not found anything beyond the trivial, and could explain the size distribution with this random site-adsorption model.

To convey this point, we have added this sentence to the main text:

“Whilst these structures catch the eye, we could not find any signature of physical significance, e.g. wall-induced phase separation. For instance, the density distribution at the wall is not bimodal (not shown). Instead the presence of trains can be explained by pure randomness.”

In conclusion, pending these two short questions, I think the manuscript is worth publishing in Nature Communications, and will likely have an impact on the more general community.

We thank the Referee for this positive interaction and their appreciation of the potential impact of our manuscript. He hope that the revised manuscripts has answered their concerns and is now suitable for publication in Nature Communications.

REVIEWERS' COMMENTS

Reviewer #2 (Remarks to the Author):

The authors have at this point answered all my comments satisfactorily. The two main issues I had related to detention time and the train structures have been either fixed or nuanced so that there is no confusion in my opinion as to what their model does and fails to do and how these structures emerge.

I think therefore the manuscript can be published as is.